# The Patient Hides the Truth, but the Computer Tomography Examination Restores It!

**DOI:** 10.3390/diagnostics12040876

**Published:** 2022-03-31

**Authors:** Liliana Dragomir, Virginia Marina, Cosmina Alina Moscu, Mihaela Anghele

**Affiliations:** 1Clinical-Medical Department, Faculty of Medicine and Pharmacy, Dunărea de Jos University of Galati, 800201 Galati, Romania; lilianadragomir2017@gmail.com (L.D.); mihaela.anghele@ugal.ro (M.A.); 2Medical Department of Occupational Health, Faculty of Medicine and Pharmacy, Dunărea de Jos University of Galati, 35 Str. Al. I. Cuza, 800201 Galati, Romania; 3Emergency Department of Hospital of Galati, 800201 Galati, Romania; cosmina_caluian@yahoo.com

**Keywords:** Fournier gangrene, anorectal trauma, intrarectal foreign object

## Abstract

In this article, we present the case of a patient who hid the real reason for which he came to the emergency room. This patient in this case is a 61-year-old man, who presented to the emergency department for symptoms relevant to respiratory disease. However, the initial diagnosis turned out to be only the “tip of the iceberg” in the evolution and treatment of this patient. Fournier gangrene is a fulminant form of infectious necrotizing fasciitis of the perineal, genital, or perianal regions, which frequently affects men, but can also occur in women and children. It is a rare but life-threatening disease. Due to potential complications, it is important to diagnose Fournier gangrene as early as possible. Although antibiotics and aggressive debridement have been widely accepted as standard treatment, the mortality rate remains high. In many cases, the doctor’s mission is much more difficult than it first seems. The challenges to which the doctor is subjected, especially in the emergency room, are immense. In establishing a diagnosis and emergency treatment, all parties involved—medical staff, on the one hand, and patients and relatives, on the other hand—must collaborate so that the outcome is a positive one. The overall conclusion of the medical team was that Fournier gangrene in the patient was the consequence of both the presence of an intrarectal foreign body, as well as excessive alcohol consumption. The case of our patient with Fournier gangrene and an intrarectal foreign body had a positive outcome due to quick, aggressive, and adequate medical and surgical management, as well as the fact that the patient had no other associated pathologies beyond alcohol “passion.” Our patient’s case is an example of the exceptional situations we sometimes encounter in medical practice.

## 1. Introduction

According to the Code of Medical Deontology of the Romanian College of Physicians “Human health is the ultimate goal of the medical act. The doctor’s obligation is to protect human physical and mental health, to alleviate suffering, to respect the life and dignity of the human person, without discrimination based on age, sex, race, ethnicity, religion, nationality, social status, political ideology or any another reason, in time of peace, as well as in time of war.” Thus, through a logical and orderly approach from the clinician in examining a patient, a correct diagnosis is established and the establishment of the necessary treatment results in healing. However, there are situations in which, with all the good intentions of the doctor, things do not go the right way, especially when patients knowingly omit anamnestic data related to the history of the current episode of the disease, leading to a misdiagnosis, making treatment difficult, or even aggravating the general condition of the patient.

Fournier’s gangrene is a fulminant form of infectious necrotizing fasciitis of the perineal, genital, or perianal regions, which frequently affects men, but can also occur in women and children [1].

Initially, Fournier’s gangrene was defined as an idiopathic entity, but careful research will indicate the source of the infection in most cases, either as perineal and genital skin infections. Anorectal or urogenital and perineal injuries, including pelvic and perineal injuries or pelvic interventions, are other causes of Fournier’s gangrene [2]. The most common foci include the gastrointestinal tract (30–50%), followed by the genitourinary tract (20–40%) and skin lesions (20%).

Usually, these are polymicrobial infections, the most common being those with streptococci, staphylococci, enterobacteria, and anaerobes.

The predisposing factors for the appearance of Fournier’s gangrene are diabetes, alcoholism, malnutrition, perineal infections, AIDS, morbid obesity, cirrhosis, leukemia, systemic lupus erythematosus, malnutrition, advanced age, and immunosuppressive therapies. Diabetes is reported to be present in 20–70% of patients with Fournier gangrene, and chronic alcoholism in 25–50% of patients [3,4].

The development and progression of gangrene are often fulminant and can quickly cause multiple organ failure and death. Due to potential complications, it is important to diagnose as early as possible. Although antibiotics and aggressive debridement have been widely accepted as standard treatment, the mortality rate remains high, being between 7% and 75% [5,6].

In this sense, both anamnesis and clinical examination have an important role in establishing a diagnosis and instituting the medical–surgical treatment as early as possible. The timing of the referral to the doctor, as well as the patient’s comorbidities, are factors on which the subsequent evolution largely depends.

## 2. Case Presentation

This patient in this case is a 61-year-old man, who presented to the emergency department for symptoms relevant to respiratory disease.

Upon admission to the emergency department: general state influenced, conscious (GCS 15 points), febrile (temperature = 38.5 °C); pale, dehydrated skin; respiratory rate = 20 breaths/minute; ventricular rate 155 beats/minute; blood pressure = 100/65 mmHg; SpO_2_ 90% in atmospheric air. Among the personal pathological antecedents, we mention: Chronic Heart Failure NYHA class II, permanent atrial fibrillation, sequelae of pulmonary tuberculosis, chronic obstructive pulmonary disease, chronic smoking, chronic alcoholism.

The anamnesis shows that the beginning of the current episode occurred approximately 7 days before the presentation, with fever and unproductive cough accompanied by a feeling of heaviness in breathing (dyspnea) with progressive character. There was no treatment for the patient’s current condition at home.

It was recommended to perform laboratory and imaging investigations, which suggested systemic inflammatory response syndrome (SIRS)as in the Table 1.

X-ray cord-lung: interstitial drawing of reticular micronodular type, intensely accentuated perihilar and bilateral infrahilar, with a tendency to bilateral hilum-basal condensation. Pulmonary hilum with enlarged projection area. Cord, aorta in normal limit (Figure 1).

Based on these data and the clinical examination, the diagnosis of bronchopneumonia was established. The patient was hospitalized in the internal medicine department, where specific treatment started immediately.

In evolution, the general condition did not improve, and about 24 h after hospitalization, the patient complained of intense pain and tenderness in the urogenital region. Locally, the skin is erythematous, edematous, cyanotic, hardy, and with blisters (Figure 2 and Figure 3).

The suspicion of Fournier’s gangrene was raised, and emergency surgery was performed, conducting necrosectomy and layered incisions up to the healthy tissue, as well as installation of minimal suprapubic cystostomy. 

Aerobic bacterial culture of secretions revealed the presence of Klebsiellosis (Enterobacter Streptomyces-resistant).

A drug sensitivity test revealed:Amikacin-resistant, Augmentin-resistant, Biseptol-resistant, Cefaclor-resistant, Cefepime-resistant, Cefoperazone-resistant, Cefotaxime-resistant, Ceftazidime-resistant, Ceftriaxone-resistant, Cefuroxime-resistant, Cefazolin-resistant, Ciprofloxacin-sensitive, Ertapenem-resistant, Gentamicin-resistant, Imipenem-sensitive, Levofloxacin-resistant, Meropenem-resistant, Moxifloxacin-resistant, Netilmicin-resistant, Ofloxacin-resistant, Tazocin-resistant, Tetracycline-sensitive, Tigecycline-resistant, Tobramycin-resistant.

The urinalysis revealed the presence of Enterococcus saprophytes > 100,000 CFU/mL, with sensitivity according to the drug sensitivity test to Fosfomycin, Nitrofurantoin, Teicoplanin, Tetracycline, and Vancomycin.

Abdominal ultrasonography performed postoperatively indicated kidneys with normal size and ultrasound structure, bladder with percutaneous cystostomy, spine with normal size and homogeneous structure, hyperechoic pancreas, homogeneous at the cephalic level, otherwise unseen, gallbladder with hyper-ultrasound density occupying 2/3 lower liver with large dimensions (left hepatic lobe 63 mm, right hepatic lobe 175 mm), homogeneous ultrasound structure, increased reflectivity, permeable vein porta, undiluted intrahepatic bile ducts; free main bile ducts.

Biologically, there was a slight decrease of inflammatory tests, as in the Table 2.

Due to the worsening general conditions, the patient was submitted to thoracic, abdominal, and pelvic computer tomography scans, which revealed:Numerous central and peripheral emphysema bubbles, with regular thin walls, located at the level of both lung fields, the largest located apically superior bilaterally;Numerous bronchiectasis with variable dimensions, some with free lumen, located in both upper lobes, in the right middle lobe and lingular segments;Pulmonary condensation areas in the middle and posterobasal segments of both lower lobes, especially on the left side (Figure 4).Bilateral mediastinal, laterotracheal lymphadenopathy (maximum axial diameter: 13 mm on the right side and 11 mm on the left side), right hilar (maximum 13 mm), and infrared (maximum 18 mm);Bilateral posterobasal pleurisy (liquid blade with a thickness of 40 mm on the right side and 41 mm on the left side) (Figure 5).No accumulation of intrapericardial fluids (Figure 6).Liver with slightly increased dimensions (craniocaudal diameter of right liver lobe, 167 mm), homogeneous structure, regular contours. Port vein in hil, 13 mm; permeable. Intrahepatic bile ducts and extrahepatic bile ducts undilated;At the rectal level, the presence of a foreign body is highlighted, with regular edges a maximum diameter of 65 mm (Figure 7 and Figure 8).

A complete and very thorough physical examination was conducted in the internal medicine department and revealed that the scrotal and anal skin was erythematous, edematous, cyanotic, and hard, with vesicles. These aspects of the examined sites made us consider the possibility of Fournier’s gangrene at an early stage.

The definite diagnosis of Fournier’s gangrene was made postoperatively.

On palpation of the area, the patient felt pain, but otherwise denied its existence.

The patient was asked if he had problems with micturition: initially, he denied it, but then said that sometimes he felt a little pain.

There is no history of urinary infections in his personal history.

Urinalysis showed mild urinary tract infection; urine color was only slightly changed; laboratory tests slightly changed; inflammatory signs present; no signs of hyperplasia or prostate cancer (PSA test normal); no diabetes mellitus.

At no time did the patient say anything about the foreign object inserted anally.

There was no evidence of mental illness in the patient.

We believe that the patient inserted the foreign object due to excessive alcohol consumption.

In this situation, surgery is performed by a mixed surgical team, comprising both urology and general surgery, intraoperatively highlighting the rupture of the grade V urethra and inside the rectal foreign body with anterior rectal wall lesion. The extraction of the foreign body (glass cup) is carried out by double approach, cystostomy, ligation of the bladder neck, minimal cystostomy, and cystography.

In our case, the operating protocol of the surgery to remove the foreign body (the glass cup) was carried out in two stages: the first stage was performed by the general surgeon, who performed an exploratory laparotomy, followed by a double abdominal and transanal approach to remove the foreign body.

The second stage was performed by the urological surgeon, who performed a cystostomy, followed by a necrosectomy up to the healthy tissue, with a collection of bile samples carried out, for culture and drug sensitivity testing.

Subsequently, the suprapubic cystostomy was performed, as the patient had a rectovesical fistula. Finally, due to soft tissue defects in the scrotum, excisional debridement with testicular relocation, as well as suturing and dressing, was performed.

Abdominal ultrasonography performed five days postoperatively revealed:

Left lobe liver 67 mm, right lobe liver 163/1557 mm, slightly increased echogenicity, microgranular structure, portal vein 11 mm, main bile duct 4 mm, undiluted intrahepatic bile ducts. Relaxed gallbladder, with 104 mm long shaft, with hyperechoic deposit, inhomogeneous in 2/3 lower, gallbladder walls 3 mm. Homogeneous, hyperechoic, and hypotrophic pancreas. Spleen with 87 mm long shaft, homogeneous. Long axis of the right kidney 119 mm, net contour, inferior posterior 12 mm. Long axis of the left kidney 90 mm, erased contour, inferior posterior 8 mm, with erasure of the corticomedullary differentiation, without dilated pathways. Invisible bladder. Free intraperitoneal fluid, in medium quantity, supra- and inframesocholic, with a partitioned appearance in the hypochondrium and left flank. Bilateral pleural fluid, in medium quantity on the left, with inhomogeneous appearance.

At 7 days, a left orchiectomy was performed, the histopathological examination concluding the nonspecific chronic orchitis, with extensive areas of sclerosis and chronic, acute vaginalitis.

From a biological point of view, a significant improvement was observed as in the Table 3.

The slow favorable postoperative evolution was burdened by the appearance of acute Enterocolitis with Clostridium difficile, for which Vancomycin was established, with the disappearance of diarrheal stools.

The pulmonary radiological examination performed 30 days after hospitalization revealed: veiling of the left lung field, bilateral accentuated lung drawing, with alveolar condensation processes, predominantly left lung; opacification of the left cost-diaphragmatic sinus, left apical pulmonary hypertransparency; increased projection area of the left hilum; mitral configured cord (Figure 9).

In dynamics, abdominal ultrasonography performed at 2 weeks postoperatively reveals: liver with normal dimensions, homogeneous structure, diffuse increased reflectivity. Permeable port vein. Undiluted intrahepatic bile ducts. The main bile duct is free. Gall bladder with hyperechoic content. Hyper-echogenic, homogeneous pancreas. Spleen with normal dimensions, homogeneous. Both kidneys with normal size and structure.

At an interval of one month after the foreign body extraction surgery, excisional debridement and secondary suturing are performed, with a good subsequent evolution.

After a 64-day hospitalization, of which more than half was in Intensive Care, although presented in toxic-septic shock, the patient was discharged in a relatively good general condition, afebrile, hemodynamically stable and respiratory, with suprapubic cystostomy, and being able to feed and mobilize.

## 3. Discussion

This article does not bring any innovation regarding the management of Fournier syndrome. Instead, our case report draws attention to the fact that the diagnosis and subsequent evolution of a patient often depend on good collaboration between patient and doctor.

In the case of our patient, concealment of the truth, the absence of subjective symptoms related to anorectal pathology, and the presence of respiratory symptoms delayed the correct management of the patient. We will never know if the patient has knowingly hidden the truth or, due to alcohol consumption, had forgotten how he got into this situation.

The behavior of this patient who tried to hide his conditions can represent an important lesson for clinical practice.

Several etiological factors for Fournier’s gangrene have been identified. Of these, urogenital trauma accounts for 25% [7].

The evolution is burdened by the rapidity of establishing the diagnosis and the surgery.

The prognosis largely depends on the time of presentation to the doctor, but also on the patient’s comorbidities.

The case we presented initially had a dragging evolution, mainly due to the lack of cooperation of a patient who did not provide correct anamnestic data. Subsequently, the evolution was favorable, due to the identification of the cause of Fournier’s gangrene, the emergency surgery, and the general treatment correctly conducted.

The patient was discharged with the following recommendations: weekly monitoring at the family doctor for cystostomy supervision; re-evaluation of the patient by the urologist for cystostomy removal in 2 months; and the inclusion of the patient in a different program to improve his quality of life, such as alcohol withdrawal and psychological counseling.

## 4. Conclusions

The overall conclusion of the medical team is that Fournier’s gangrene was the consequence of the presence of an intrarectal foreign body and excessive alcohol consumption.

In cases such as these, the doctor’s mission is much more difficult than it first seems. The challenges to which the doctor is subjected, especially in the emergency room, are immense. In establishing a diagnosis and emergency treatment, all parties involved—medical staff, on the one hand, and patients and relatives, on the other hand—must collaborate so that the outcome is a positive one.

The case of our patient with Fournier’s gangrene and an intrarectal foreign body had a positive outcome, due to the quick, aggressive, and adequate medical and surgical management, as well as the fact that the patient had no other associated pathologies, beyond alcohol “passion.” Our patient’s case is an example of the exceptional situations we sometimes encounter in medical practice.

## Figures and Tables

**Figure 1 diagnostics-12-00876-f001:**
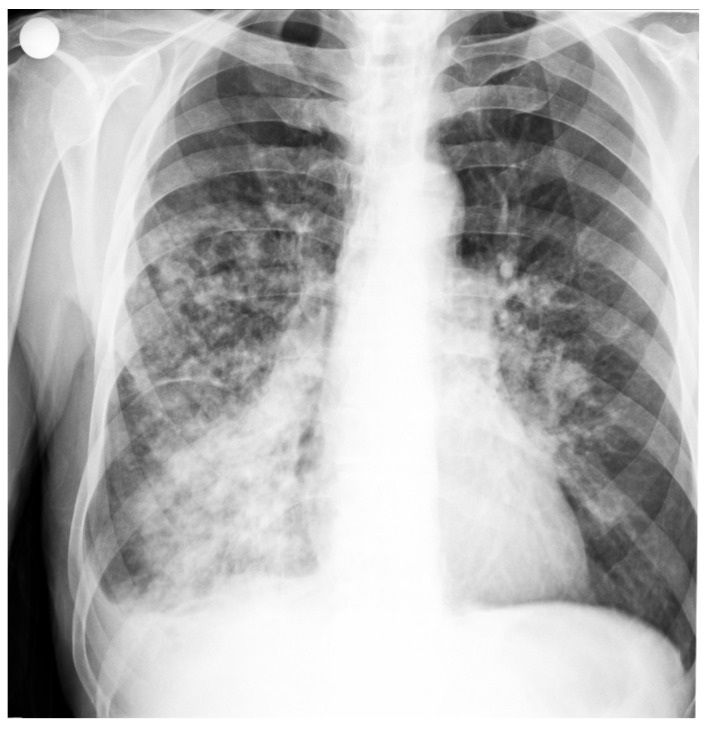
Patient’s Rx cord-lung: interstitial drawing of reticular micronodular type, intensely accentuated perihilar and bilateral infrahilar, with a tendency to bilateral hilum-basal condensation.

**Figure 2 diagnostics-12-00876-f002:**
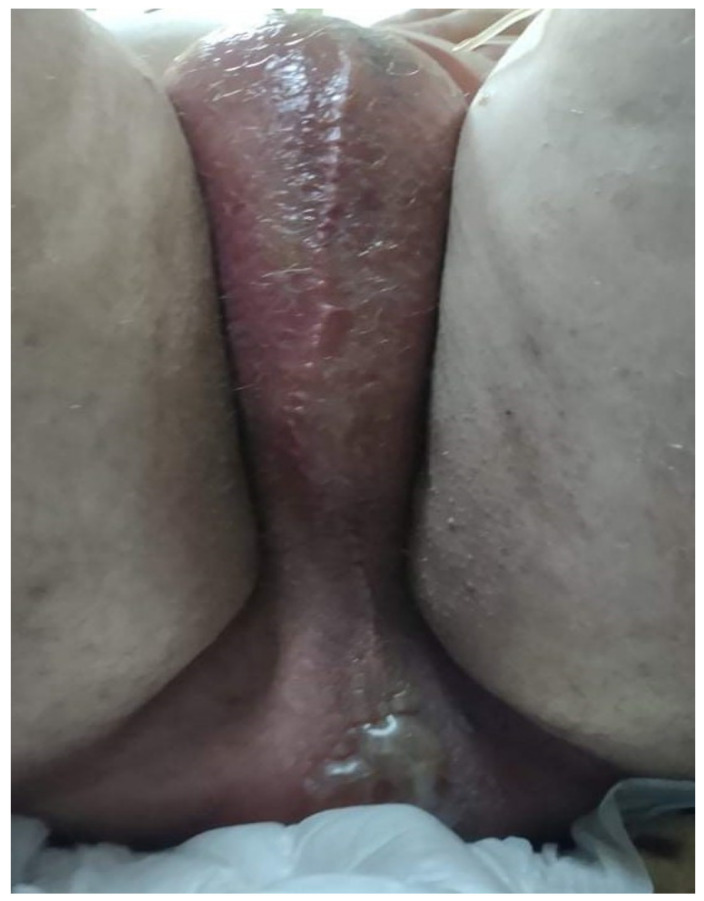
Patient’s skin: the aspect is erythematous, edematous, cyanotic, and hardy, with blisters.

**Figure 3 diagnostics-12-00876-f003:**
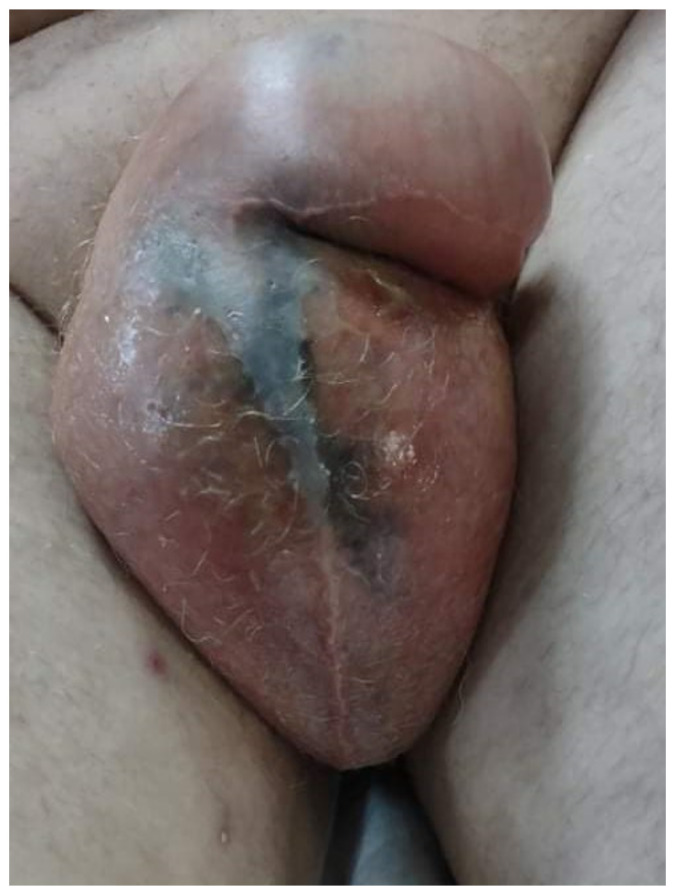
Patient’s Fournier gangrene.

**Figure 4 diagnostics-12-00876-f004:**
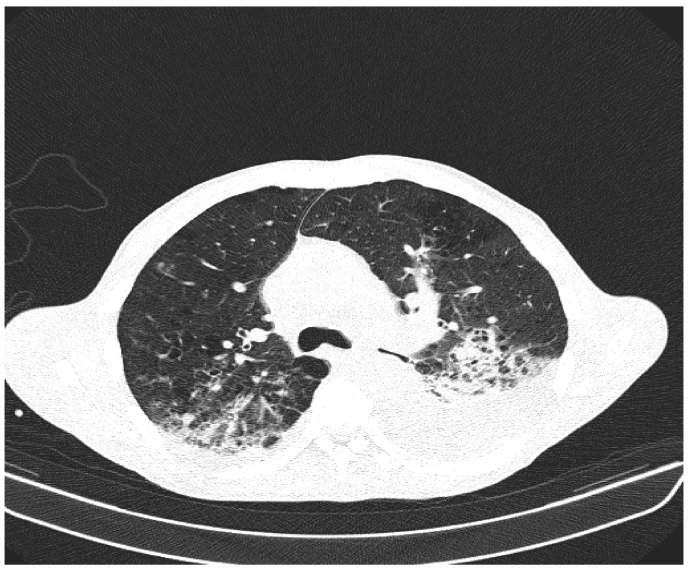
Patient’s computer tomography: pulmonary condensation areas in the middle and posterior-basal segments of both lower lobes, especially on the left side.

**Figure 5 diagnostics-12-00876-f005:**
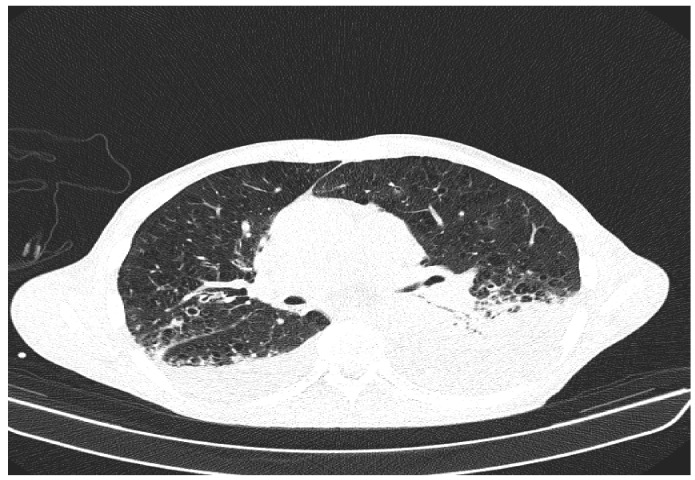
Patient’s computer tomography: bilateral posterior-basal pleurisy.

**Figure 6 diagnostics-12-00876-f006:**
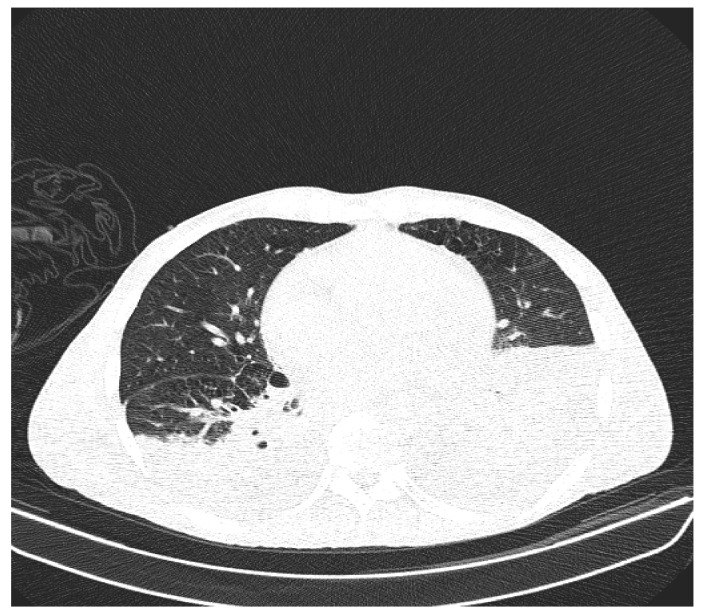
Patient’s computer tomography: no fluid accumulation intrapericardial.

**Figure 7 diagnostics-12-00876-f007:**
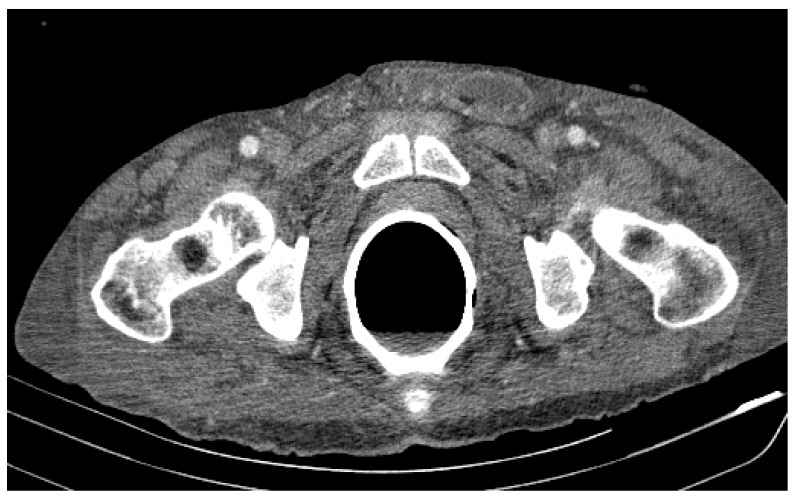
Patient’s computer tomography: the presence of a foreign body is highlighted.

**Figure 8 diagnostics-12-00876-f008:**
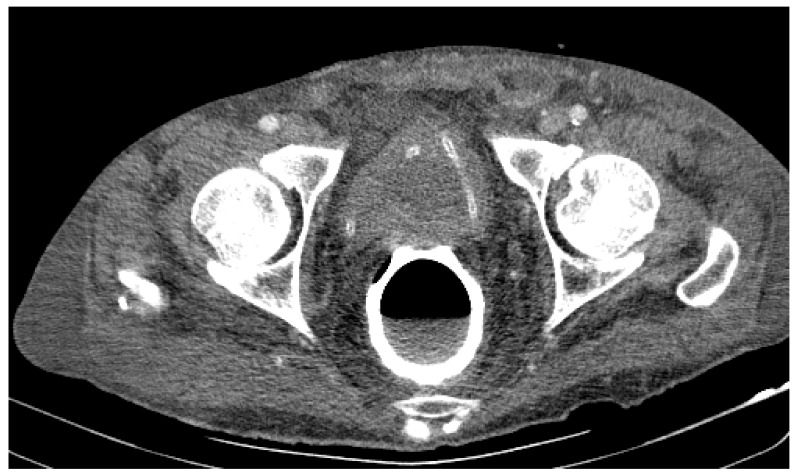
Patient’s computer tomography: the presence of a foreign body is highlighted, with regular edges and a maximum diameter of 65 mm.

**Figure 9 diagnostics-12-00876-f009:**
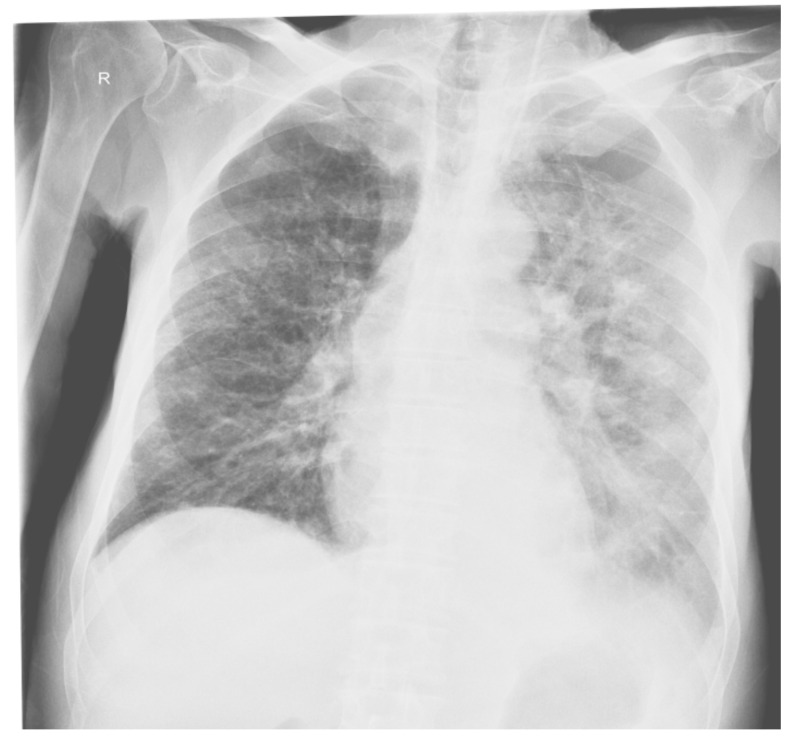
Patient’s X-ray (performed 30 days after hospitalization): bilateral accentuated lung drawing, with alveolar condensation processes, predominantly left lung; opacification of the left cost-diaphragmatic sinus, left apical pulmonary hypertransparency. ‘R’= right.

**Table 1 diagnostics-12-00876-t001:** Patient’s laboratory investigations (systemic inflammatory tests).

Parameter	Patient’s Value
WBC	38.31 × 10^3^/µL
Neutrophil	88.10%
C-Reactive Protein	117 mg/dL
Sedimentation rate of Hematite	25 mm/hour
Fibrinogen	744 mg/dL

**Table 2 diagnostics-12-00876-t002:** Patient’s value of re-evaluation(decrease of inflammatory tests).

Parameter	Patient’s Value
WBC	20.88 × 10^3^/µL
Neutrophil	90%
Fibrinogen	438.8 mg/dL
C-reactive protein	80 mg/dL

**Table 3 diagnostics-12-00876-t003:** Patient’s laboratory investigations after orchiectomy( biological improuvement).

Parameter	Patient’s Value
WBC	8.86 10^3^/µL
Neutrophil	83%
C-reactive protein	29 mg/dL

## Data Availability

Informed consent was obtained from all subjects involved in the study.

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
