# Peer review of "The Patient Hides the Truth, but the Computer Tomography Examination Restores It!"

_diagnostics, 2022, doi:10.3390/diagnostics12040876_

Round 1

Reviewer 1 Report

Authors reported a case report of a man presenting pneumonia and Fournier gangrene due to a non-reported intrarectal foreign body.

The study aims to demonstrate that timing of the referral to the doctor and the patient's comorbidities are factors on which the subsequent evolution of Fournier gangrene will largely depend.

Author Response

Authors reported a case report of a man presenting pneumonia and Fournier gangrene due to a non-reported intrarectal foreign body.

At no time did the patient say anything about the foreign object inserted anally.

Only the CT scan revealed the existence of a foreign body.

The study aims to demonstrate that timing of the referral to the doctor and the patient's comorbidities are factors on which the subsequent evolution of Fournier gangrene will largely depend.

 The overall conclusion of the medical team was that gangrene Fournier was the consequence of the presence of

intra-rectal foreign body and excessive alcohol consumption.

The patient had no other associated pathologies.

Reviewer 2 Report

The authors present an interesting case focusing on Fournier´s Gangrene. There are minor changes, that can be performed:

  • Please enrich the patient information during initial presentation

Thank you

Author Response

The authors present an interesting case focusing on Fournier´s Gangrene. There are minor changes, that can be performed:

  • Please enrich the patient information during initial presentation
  • Following the complete and very thorough physical examination that was done in the internal medicine department, the following were revealed: as for the scrotal and anal skin, it was erythematous, edematous, cyanotic, hard, with vesicles. These aspects of the examined sites made us think of to a possible Fournier gangrene at an early stage.
  • The definite diagnosis of Fournier gangrene was made postoperatively.
  • On palpation of the area, the patient felt pain, but otherwise denied its existence.
  • He was asked if he had problems with micturition: initially he denied it and then said that sometimes he felt a little pain.
  • There is no history of urinary infections in his personal history.
  • - Urinalysis showed mild urinary tract infection; urine color was only slightly changed; laboratory tests slightly changed; inflammatory signs present; no signs of hyperplasia or prostate cancer (PSA Test normal); no diabetes mellitus.

Reviewer 3 Report

The case does not add new informations about how to manage Fourniere's Gangrene. However it can be reported considering the behaviour of this patient who tried to hide his condition and this fact can represent an important lesson for clinical practice. 

I understand that the initial symptoms evaluation induced the colleagues to consider pulmonary acute disease, however a complete and well done physical examination should have revealed also the urological problem. 

Therefore I think that the title of this manuscript may be considered correct, but I think also that in the final considerations you have also to highlight that patients shold be completely examined. This was a 61-years old man. Urinary tract infection, prostatic problems (including prostatic hyperplasia / cancer) can affect men over 50 yrs. Therefore, a complete phyisical examination must be performed. In this case it appears to me that a complete physical examination has not been performed neither in emergency room (when the patient arrived to the hospital) nor when patient was accepted in internal medicine department. The patient was visited only when he reported pain in anorectal region. This is not acceptable. 

Don't You perform a urine examination (i.e. urinary stick) during the first triage in the Emergency Department ? Please comment. 

I think the case could be more interesting if You could provide intraoperative pictures showing the glass inside the rectum, the lesion of rectal wall and of urethra. You reported a type 5 urethral lesion, please specify complete / incomplete (specfying to which classification You refer) and if is possible provide images. 

Generally patients suffering from Fourniere's Gangrene need early management of scrotal / testicular infection. Probably I did not understand the steps of your surgical treatment, and therefore I would ask You if You initially put a drain at scrotal level to remove the scrotal abscess and why you did not perform orchidectomy / scrotal surgery / management at the same time of abdominal procedure but only 7 days after (it is only to understantd). 

Usually patients suffering from Fourniere's Gangrene need multiple subcutaneous suprapubic and abdominal drainage to allow the removal of infectious secretions along the fascia. Was this necessary in your case ? Please comment. 

Some Authors report the use of Vacuum Assisted Closure (VAC) therapy to accelerate the restorgin of skin integrity. Did You use this approach in your case ? Was it not necessary ? Please comment. 

Which is the plan fot his patient ? Will he continue his life carrying a cystostomy ? Did You consider to re-evaluate the patient for a different program thus to improve his quality of life ? 

Author Response

The case does not add new informations about how to manage Fourniere's Gangrene. However it can be reported considering the behaviour of this patient who tried to hide his condition and this fact can represent an important lesson for clinical practice. 

I understand that the initial symptoms evaluation induced the colleagues to consider pulmonary acute disease, however a complete and well done physical examination should have revealed also the urological problem. 

Therefore I think that the title of this manuscript may be considered correct, but I think also that in the final considerations you have also to highlight that patients shold be completely examined. This was a 61-years old man. Urinary tract infection, prostatic problems (including prostatic hyperplasia / cancer) can affect men over 50 yrs. Therefore, a complete phyisical examination must be performed. In this case it appears to me that a complete physical examination has not been performed neither in emergency room (when the patient arrived to the hospital) nor when patient was accepted in internal medicine department. The patient was visited only when he reported pain in anorectal region. This is not acceptable. 

Don't You perform a urine examination (i.e. urinary stick) during the first triage in the Emergency Department ? Please comment. 

During the first triage in the Emergency Department the urine examination it is not made.

(hospital protocol does not provide for this examination in emergency department)

I think the case could be more interesting if You could provide intraoperative pictures showing the glass inside the rectum, the lesion of rectal wall and of urethra. You reported a type 5 urethral lesion, please specify complete / incomplete (specfying to which classification You refer) and if is possible provide images. 

Generally patients suffering from Fourniere's Gangrene need early management of scrotal / testicular infection. Probably I did not understand the steps of your surgical treatment, and therefore I would ask You if You initially put a drain at scrotal level to remove the scrotal abscess and why you did not perform orchidectomy / scrotal surgery / management at the same time of abdominal procedure but only 7 days after (it is only to understantd). 

Usually patients suffering from Fourniere's Gangrene need multiple subcutaneous suprapubic and abdominal drainage to allow the removal of infectious secretions along the fascia. Was this necessary in your case ? Please comment. 

For our case was not necessary because Fourniere's Gangrene was at an early stage and the lesions were not advanced.

Some Authors report the use of Vacuum Assisted Closure (VAC) therapy to accelerate the restorgin of skin integrity. Did You use this approach in your case ? Was it not necessary ? Please comment. 

VAC was not necessary in our case ! The decision belonged to the urologist.

Which is the plan fot his patient ? Will he continue his life carrying a cystostomy ? Did You consider to re-evaluate the patient for a different program thus to improve his quality of life ? 

 The patient was discharged with the following recommendations:

-weekly monitoring at the family doctor for cystostomy supervision

-re- evaluate the patient by the urologist for cystostomy removal in 2 months;

-including the patient for a different program thus improve his quality of life such as: alcohol withdrawal and psychological counselling.

Round 2

Reviewer 1 Report

The paper is ok for publication.

Author Response

Thank you very much.

Reviewer 3 Report

Dear Authors, thank'you for the changes you made on the paper. However, I still found some mistakes that I'd like You're going to improve. Here my comments. 

1) Usually the Abstract resumes what the manuscript reports. This is not the case. I find the abstact quite philosophical. I suggest You to be more concrete and to insert some data related to case presentation. 

2) Still hard english review is recommended. There are some errors who have to be corrected. 

3) Reference are reported in brackets. However, sometimes the brackets are inserted before the dot, sometimes after. Please follow the Journal guidelines. 

4) Page 2: you cannot report the predisposing factors to Fournier's Gangrene including the word "etc". This is a scientific paper: readers cannot imagine other predisposing factors that you are not citing. Please indicate all the predisposing factors. If the list is long You can introduce a table in the paper. 

5) At the end of "Introduction" you report that clinical examination has important role to diagnose this condition. However, a full clinical examination was not done in this case at emergency department. 

6) I think that first and second sentences of the "Case presentation" are quite repetitive. Furthermore You use the word "present" too many times ("we present", "presenting", again "we present", then "who presented"); please improve the grammatical form of these parts.

7) In the paragraph "Case presentation", aftet "atmospheric air" You have to put a dot. 

8) Change the word "tubercollosis" with the correct form "tubercolosis".

9) The sentence "the internal medicine department is hospitalized" has no sense. Probably the correct sentence is "the patients has been hospitalized at the internal medicine department". 

10) I think that all surgical treatment and follow-up have to be reported in the case presentation and NOT in the "discussion" paragraph. 

11) I do not clearly understand the surgery, because You report "cystostomy" and immediately after "minimal cystostomy". Probably is a repetition. 

12) The sentence: "During the second stage is performed by the urological surgeon" has to be changed into: "The second stage has been performed by the urological surgeon".

13) The sentence: "No photographs were taken during sugery" is quite embarassing if you want to publish a manuscript as a case presentation which needs ICONOGRAPHY. Do not You agree ?

14) Conclusion must be shortened. Too long. The conclusion is to report THE FINAL MESSAGE. Not more.  

Actually, I found that this manuscript is getting worse. 

Please make all the changes I asked.

Make a serious english review BEFORE submitting the final version. 

Thank'you very much.  

Author Response

Dear Authors, thank'you for the changes you made on the paper. However, I still found some mistakes that I'd like You're going to improve. Here my comments. 

  • Usually the Abstract resumes what the manuscript reports. This is not the case. I find the abstact quite philosophical. I suggest You to be more concrete and to insert some data related to case presentation. 

You're right. I changed it :

Abstract:

In this article we present the case of a patient who hid the real reason for that he came to the emergency room.

 This patient is a 61-year-old man who presented to the Emergency Department for symptoms relevant to respiratory disease. But the initial diagnosis turned out to be just the "tip of the iceberg" in the diagnosis, evolution and treatment of this patient.

Fournier gangrene is a fulminant form of infectious necrotizing fasciitis of the perineal, genital, or perianal regions, which frequently affects men, but can also occur in women and children. [1] About Fournier gangrene we can say that it is a rare but life-threatening disease. Due to potential complications, it is important to diagnose as early as possible. Although antibiotics and aggressive debridement have been widely accepted as standard treatment, the mortality rate remains high. We can say that the doctor's mission is much more difficult than it seems. The challenges to which the doctor is subjected, especially in the emergency room, are great, and in establishing a diagnosis and establishing an emergency treatment must collaborate all parties involved, medical staff on the one hand and patients and relatives, on the other hand, so that the outcome is a happy one.

The overall conclusion of the medical team was that gangrene Fournier was the consequence of the presence of intra-rectal foreign body and excessive alcohol consumption.

The case of our patient with Fournier's gangrene and intra-rectal foreign body had a happy ending due to the quick, aggressive and adequate medical and surgical management and the fact that the patient had no other associated pathologies but only alcohol "passion". Our patient's case is one of those rare cases we met in medical practice.

  • Still hard english review is recommended. There are some errors who have to be corrected. 

If I have another errors I hope that the journal's proofreading department will make the final edits to the manuscript.

  • Reference are reported in brackets. However, sometimes the brackets are inserted before the dot, sometimes after. Please follow the Journal guidelines. 

Ok I have done.

  • Page 2: you cannot report the predisposing factors to Fournier's Gangrene including the word "etc". This is a scientific paper: readers cannot imagine other predisposing factors that you are not citing. Please indicate all the predisposing factors. If the list is long You can introduce a table in the paper. 

All predisposing factors for Fournier’s gangrene are :

diabetes mellitus - in more than 60% of cases there is an association; alcoholism; morbid obesity; cirrhosis; leukaemia; systemic lupus erythematosus; HIV infection; malnutrition; advanced age; immunosuppressive therapies

  • At the end of "Introduction" you report that clinical examination has important role to diagnose this condition. However, a full clinical examination was not done in this case at emergency department. 

A full clinical examination was done in this case at emergency department but the patient t did not say anything about the foreign body.

He denied all along that he had introduced his foreign body.

He told us he didn't know why he had scrotal skin like that.

6) I think that first and seconde sentence of the "Case presentation" are quite repetitive. Furthermore You use the word "present" too many times ("we present", "presenting", again "we present", then "who presented"); please improve the grammatical form of these parts.

I corrected it.

7) In the paragraph "Case presentation", aftet "atmospheric air" You have to put a dot. 

I corrected it.

8) Change the word "tubercollosis" with the correct form "tubercolosis".

I corrected it « tuberculosis »

9) The sentence "the internal medicine department is hospitalized" has no sense. Probably the correct sentence is "the patients has been hospitalized at the internal medicine department". 

I corrected it.

10) I think that all surgical treatment and follow-up have to be reported in the case presentation and NOT in the "discussion" paragraph. 

I corrected it.

11) I do not clearly understand the surgery, because You report "cystostomy" and immediately after "minimal cystostomy". Probably is a repetition. 

I corrected it.

12) The sentence: "During the second stage is performed by the urological surgeon" has to be changed into: "The second stage has been performed by the urological surgeon".

I corrected it

13) The sentence: "No photographs were taken during sugery" is quite embarassing if you want to publish a manuscript as a case presentation which needs ICONOGRAPHY. Do not You agree ?

Ok.

14) Conclusion must be shortened. Too long. The conclusion is to report THE FINAL MESSAGE. Not more.  

I’ve  changed  it.

Actually, I found that this manuscript is getting worse. 

Please make all the changes I asked.

Make a serious english review BEFORE submitting the final version. 

Thank'you very much.  

Round 3

Reviewer 3 Report

Abstract: 1) the sentence "but the initial diagnosis turned out to be just the tip of the iceberg in the diagnosis" must be changed (diagnosis is repeated two times in the same sentence); 2) please remove words as "we can say"; it is not professional and is not used in Literature; 3) you should replace "great" in association to "challenges", is not grammatically correct (usually "great" has a positive meaning,then in this case its use as attribute is not correct); 4) remove the words "happy ending" and replace with "favorable outcome" or something similar that results more appropriate for a scientific paper.

Introduction:  please remove "we can say that" in the final sentence. 

Case presentations: 1) please remove the "s" and rewrite "case presentation"; 2) I'd like to remove the firs sentence cause you reveal the conclusion of the case; the same for the second sentence: start with the patient arrival at the Emergency Department; 3) "the anamnesis shows that the beginning of the current episode began" has to be modified (beginning followed by began, is grammatically not nice); 4) based on thiese data, use "basing on these data ..."; 5) You should represent the data of antibiogram in a table, is too much long the description in the text; 6) after table 2 please write as follows: "Due to the worsening general conditions, the patient has been submitted to thoracic, abdominal and pelvic CT which revealed ...", then list the CT findings entirely removing the sentence "Compute tomography of the thoarx...... (is a repetition); 7) please remove capital letters at the beginning from the list of CT findings (i.e. change Numerous to numerous, Pulmonary to pulmonary) .......; 8) the sentence "at no time did the patient say anything...." is grammatically not correct, please change; the same for the followgin sentence, please correct; 9) there are other errors (i.e. "surgery is performed in the mixed team" has to be modified using "by a mixed surgical team". 

Discussions: please remove the "s" and rewrite: "discussion"; I think the discussion is quite short and it does not provide suggestions to overcome the difficulties you found managing with this patient. 

Therefore I suggest to: 

  • Make a serious english review. I already suggested you to do this. If you cannot do by yourself, please refer to "official services" that provide this contribute (San Francisco Edit or similar).
  • The manuscript needs to be better organized to improve reader's comprehension, the extension should be reduced cause is too long for a case presentation. 

Author Response

Please see the attachment, thank you very much.

Round 4

Reviewer 3 Report

I asked a extended english revision and some changes in the previous revision that have not been done. 
Please refer to previous revision.